Testing ‘Proportion of Females Calving’ as an indicator for population-level reproductive performance for black rhinoceros (Diceros bicornis)

Law Peter R. prldb55@gmail.com 1
Fike Brad 2
1 Centre for African Conservation Ecology, Department of Zoology, Nelson Mandela University , Port Elizabeth , South Africa
2 Port Alfred , South Africa
Hedrick Ann
Electronic publication date: 2018 Aug 15
Publication date: 2018
Volume: 6
Electronic Location ID: e5430
Received 2018 Feb 19; Accepted 2018 Jul 21
Copyright: ©2018 Law and Fike
Copyright year: 2018
Copyright holder: Law and Fike
License: This is an open access article distributed under the terms of the Creative Commons Attribution License, which permits unrestricted use, distribution, reproduction and adaptation in any medium and for any purpose provided that it is properly attributed. For attribution, the original author(s), title, publication source (PeerJ) and either DOI or URL of the article must be cited.
License URL: https://creativecommons.org/licenses/by/4.0/

Keywords: Black rhinoceros, Megaherbivores, Population reproductive performance

Funding: The authors received no funding for this work.

==============================
The proportion of females calving (PFC) each year has been employed as an indicator of population reproductive performance in ungulates, especially for species that breed annually, because it requires less detailed population data than inter-birthing intervals and age at first reproduction. For asynchronous breeders with inter-birthing intervals longer than a year such as megaherbivores, however, it is unclear how much annual variation in PFC is expected and whether false signals of density feedback or environmental influence might result from analyzing PFC data. We used census data from a well studied, closed, expanding population of black rhinoceros (Diceros bicornis) to study annual variation in PFC over 22 years. Our analysis of PFC data yielded no false signals of density feedback but weak evidence for an unexpected influence of rainfall. The PFC data exhibited considerable variation, which we attribute to autocorrelation in the time series of PFC data, ‘demographic-founding effects’, changes in stage structure, and demographic stochasticity, some of which the modelling of PFC appears to confuse with an influence of rainfall. We expect such variation to be common in introduced populations and to persist for some years, complicating the interpretation of PFC, though moving averages of PFC can help if employed cautiously. While our analysis does not undermine the possible utility of PFC, the analysis and interpretation of PFC values require care.

Introduction

Studying megaherbivore population dynamics is challenging due to the longevity of individuals. The resources required to identify and monitor individuals over long periods of time may be costly, involving technological challenges and long-term commitments from dedicated personnel, while the populations themselves are open to unnatural disturbances (poaching). Both intrinsic ecological interest and conservation demands, especially for rhinoceros (IUCN red list: Ceratotherium simum near threatened, Rhinoceros unicornis vulnerable, and Diceros bicornis, Dicerorhinus sumatrensis and Rhinoceros sondaicus all critically endangered), nevertheless make such studies important. In the absence of demographic histories of individuals, practical measures that might reflect population performance could be valuable.

For ungulates species with synchronized annual breeding, the proportion of adult females calving (hereafter PFC) each breeding season provides a measure of population reproductive performance (Fryxell, 1987). Imperfect detection can undermine the reliability of counts estimating PFC (Bonenfant et al., 2005), but that is not our concern here. Rather, we focus on the interpretation and utility of PFC, when accurately estimated, for megaherbivores that are asynchronized, non-seasonal breeders with inter-birthing intervals longer than a year. Observation must occur over the entire year due to the lack of a birthing season and only a fraction of mature females will be available to breed in any given year, inducing autocorrelations in a time series of PFC data. Moreover, annual variation in PFC may not reflect drivers of population performance, such as density feedback and environmental influence. Instead, annual fluctuations in PFC may be more sensitive to demographic stochasticity and ‘demographic-founding effects’, i.e., demographic consequences of the distribution of sex and (st)age amongst the introduced individuals founding reintroduced populations, especially as many extant populations of rhinoceros, in particular, are small and recently reintroduced. Nevertheless, one of Du Toit’s (2001) benchmarks of black rhinoceros population performance in the wild is ‘% of cows with calf of that year’ (<29%, very poor–poor; 29–33%, poor–mod; 33–40% mod–good; >40% good–excellent). PFC has also been exploited in demographic studies of black rhinoceros populations (Hrabar & Du Toit, 2005; Okita-Ouma, 2014).

Our aim in this paper is to investigate the performance of PFC as a measure of population-level reproductive performance by analyzing PFC for a population of black rhinoceros (Diceros bicornis minor) for which complete individual histories were known (Fike, 2011). Previous study of the demography of this population based on the complete individual histories (Law, Fike & Lent, 2013; Law, Fike & Lent, 2014; Law, Fike & Lent, 2015) permitted us to frame expectations of PFC if it is to provide a reliable measure of population-level reproductive performance and to test those expectations. We studied mortality, age at first reproduction, and inter-birthing intervals in Law, Fike & Lent (2013), and birth sex in Law, Fike & Lent (2014). Through modelling, the only evidence of density feedback we found on these demographic parameters was increasing age at first reproduction with increasing population size. We found no evidence for an influence of: total rainfall over 15 (or 24) months prior to the first birth on age at first reproduction; total rainfall during an inter-birthing interval or the six months prior to the beginning of an inter-birthing interval on the inter-birthing interval; total rainfall during gestation or total rainfall during the periods of 7, 12, or 24 months prior to conception on birth sex. Neither maternal identity nor age, nor the sex of the calf that initiated an inter-birthing interval, explained variation in inter-birthing intervals, while variation in birth sex was not accounted for by maternal identity or age, year of conception, or population adult sex ratio. Yet variation in inter-birthing intervals and birth sex persisted throughout the 22 years of the study. On the basis of these results, we conjectured that variation was due, at least in part, to demographic stochasticity. In Law, Fike & Lent (2015), we fitted semi-annual population census counts to various models of population dynamics expected to be suitable for megaherbivores (Owen-Smith, 2010). A model of exponential growth with intrinsic annual rate of growth estimated as 0.102 ± 0.017 was unambiguously the best model fit to the data. Using this scalar exponential model and the method of Lande et al. (2003:19–20) (see also Morris & Doak, 2002:127–128) and also a more detailed stage-based matrix model of the population with the method of Engen et al. (2005), we obtained estimates of demographic stochasticity that supported our earlier contention of a role for demographic stochasticity in explaining variation in birth numbers and sexes throughout the study period.

Our previous results enabled us to interpret the population-level PFC values and assess their utility. In particular, we found only a subtle signal of density feedback on our study population and no signal of environmental influence, so we can test whether analyses of PFC data can falsely suggest either influence. During the phase of exponential growth, which is expected to be prolonged for a population of megaherbivores (McCullough, 1992), annual variation in PFC values can reflect demographic stochasticity and demographic-founding effects as life stages move towards stable distribution. By quantifying this variation for our study population, we provide a measure of the background variation against which responses to density feedback and environmental influence, if present, need stand out. For our study population, the mean inter-birthing interval was 29.0 ±  0.9 months, (n = 77), so one might expect about 1∕2.5 = 0.4 of already adult females to calve each year in an exponentially growing population in good habitat, not accounting for instances of first calving during the year. We predict variation in PFC about 0.4 for our study population. We also examined moving averages of PFC values.

We expect there will be fluctuations in PFC due to autocorrelations and stochasticity, especially during phases of growth at small population size. Our previous studies have shown that, for our study population, population density and rainfall do not explain variation in individual-level indicators of demography such as inter-birthing intervals, age at first reproduction, and birth sex. We therefore hypothesize that if PFC is a reliable measure of population-level reproductive, analyses of our PFC data with covariates of population density and rainfall will not implicate these covariates as meaningful explanations of variation in PFC. We are able to test this hypothesis with the data of our study population given our previous detailed demographic study of its individual-level demography.

Materials and Methods

The study site is managed by the Eastern Cape Parks and Tourism Agency (ECPTA). The ECPTA is mandated by its enabling act to monitor black rhinoceros under its care and to conduct research in aid of their management (D Peinke, ECPTA, pers. comm., 2018). Treatment of individual black rhinoceros for the purposes of management and monitoring employs professional practices, by qualified personnel, based on decades of experience (e.g. Morkel & Kennedy-Benson, 2007). In particular, the individuals of the study population were ear notched to meet the mandated monitoring goals, in a responsible manner reflecting both the ethical treatment of animals (Sikes et al., 2011) and the high value of individuals of this critically endangered species. Monitoring itself was conducted by aerial surveys using a microlight aircraft, observations by ground patrols, and camera traps. These procedures were conducted so as to minimally interfere with the subjects. Our research merely took advantage of the data accumulated by this monitoring. Field work was therefore not a part of this study and no permits were required.

The study population was a closed population 1986–2008, apart from several introductions (totaling 23 individuals that survived introduction), which ceased at the end of 1997, and the removal of five subadults in 2006, the latter having negligible effect on demography prior to 2009. Introduced into a well defined area of 220 km2 within the Great Fish River Reserve in the Eastern Cape, South Africa, the population grew monotonically from an initial release of four animals to 110 individuals by the end of 2008 (see Fig. 1 and Table 1) with the number of female adults increasing monotonically from 1 to 29. The study site is considered excellent black rhinoceros habitat (Van Lieverloo et al., 2009; Fike, 2011). Further details may be found in Fike (2011) and Law, Fike & Lent (2013), Law, Fike & Lent (2014), Law, Fike & Lent (2015). All rhinoceroses were individually identifiable and we possessed complete individual histories 1986–2008 from which we computed exact PFC values 1987–2008 as follows. We employed the stage-based definition of ‘adult’ female of Law & Linklater (2014), reflecting the biology and life history of the species, used throughout our studies: a female is adult if she has calved or reached the age of seven years without calving. For a given year, we counted any individual that was alive as an adult female during part of that year. PFC was then the number of births during the year divided by that count.

Figure 1 Plot of PFC (•) along with population size (∘) for the study population.

Table 1 Demographic history of the study population.

Year	PFC	Female calves	Female subadults	Female adults	Male calves	Male subadults	Male adults	
1986	na	0	1 (import)	1 (import)	0	0	1 (import)	
1987	0	0	1	1	0	0	1	
1988	1	1	1	1	0	0	1	
1989	0	1	3 (2 imports)	1	0	1 (import)	1	
1990	0.3333	2	2	3 (1 import)	0	2 (1 import)	1	
1991	0.5	2	1	4	1	2	1	
1992	0.5	3	2 (1 import)	4	2	2	1 (import)	
1993	0.25	2	4	4	2	0	3	
1994	0.8	4	4	5	1	2	3	
1995	0.3333	3	4	6	3	2	3	
1996	0.125	3	2	8	4	2	3	
1997	0.5	4	12 (7 imports)	8	2	9 (5 imports)	4 (1 import)	
1998	0.4444	5	11	8	3	11	4	
1999	0.4444	6	13	9	3	10	5	
2000	0.0833	7	9	12	3	9	6	
2001	0.6471	6	9	17	7	12	6	
2002	0.1667	7	9	18	7	8	10	
2003	0.6	12	10	20	7	7	14	
2004	0.3478	13	11	23	7	8	15	
2005	0.4	12	13	24	10	10	15	
2006	0.5556	12	14 (4 exports)	27	14	9 (1 export)	17	
2007	0.2593	11	16	27	15	13	16	
2008	0.4483	14	19	29	13	19	16	

We first applied the standard arcsine-square-root transformation to PFC as is customary for proportions. We also calculated a modified version that better normalizes proportions close to zero and one (Zar, 1999 equation 13.8), see Fig. 2. As the results did not depend qualitatively on the choice of transformation, we only report results for the modified transformation, which we denote transPFC. As physical covariates we employed: population density at the beginning of the year for which PFC was computed (hereafter ‘density’); a lagged version of density, computed 12 months prior to that just defined, was highly correlated with density (0.99, reflecting the monotonic growth in population size) and was discarded as uninformative; total rainfall for the calendar year for which PFC was computed (‘rain’); and total rainfall for the prior year (‘rain1’). The two rain measures were only negligibly correlated (−0.06). The predictors were mean centred and standardized by dividing by their standard deviations.

Figure 2 The raw data PFC (•), its arcsine-square root transform (∘) and the modified version of Zar (1999; equ. 13.8) transPFC(▴).

Both transformations preserve the pattern of variation in the raw data but transPFC is more faithful than the conventional arcsine transform when raw data values are close to one.

Given that our data form a time series in which we expect autocorrelations and for which we also need to examine the possible influence of covariates, autoregressive (AR) modelling provides a suitable approach to analyzing our data. AR modelling has the further advantage of being based on maximum likelihood methods, which permits the use of the Akaike information-criterion method of model ranking and selection, which has many advantages over null hypothesis testing (Burnham & Anderson, 2002). We first examined (partial) autocorrelations to determine the order of the AR model and also the window for constructing moving averages of the raw data. We constructed AR models using the R package MARSS (Holmes, Ward & Scheuerell, 2014) with transPFC as response. We used the second-order correction AIC c (Burnham & Anderson, 2002) to rank all 25 = 32 models with different combinations of covariates. We performed model averaging over the models in this ranking whose Akaike weights summed to 0.95 and also computed the relative importance of the covariates, i.e., the sum of Akaike weights over the models in which a variable appears (Burnham & Anderson, 2002).

Results

From Fig. 3, autocorrelations do not suggest any cyclic behaviour in the time series. The most important partial autocorrelations were for lag one and lag two so we chose our global autoregressive model to be AR(2). With transPFCi denoting the value of transPFC i years prior to the response, our global model was: transPFC=b0+b1transPFC1+b2transPFC2+c1density+c2rain+c3rain1+ε

with εN(0, σ2). The global model explained 46% of the variance in the response variable, a respectable amount for an ecological model per Møller & Jennions (2002).

Figure 3 Autocorrelations (ACF; A) and partial autocorrelations (partialACF; B) for the time series of transformed PFC values, transPFC.

The plots were obtained using the R functions acf and acf(p), respectively. Horizontal dashed lines indicate 95% significance levels. Only the autocorrelation of the first lag reaches that significance level but the partial autocorrelation of the second lag only negligibly fails to do so.

The relative importances of the variables were: 0.9353, transPFC1; 0.4971, transPFC2; 0.2807, rain; 0.1609, rain1; 0.1531, density. The top 14 models, accounting for 95% of cumulative Akaike weights, appear in Table 2. The model average (Burnham & Anderson, 2002:152), over these 14 models, of the coefficients and SEs of transPFC1, transPFC2 and rain were −0.55 ±  0.21, −0.16 ± 0.20, and −0.012 ± 0.021, respectively.

Table 2 Model rankings for AR(2) models for which the cumulative Akaike weights sum to 0.95.

The covariates in these models are: TransPFC1 and transPFC2 are the one-step and two-step lags of the transformed PFC values transPFC; rain, the total rainfall for the calendar year for which PFC was computed; rain1, the total rainfall for the prior year; and density, the population density at the beginning of the year for which PFC was computed.

Model	ΔAICc	Model	ΔAICc	
transPFC1 + transPFC2	0	transPFC1 + transPFC2+density	3.47	
transPFC1	0.07	transPFC1 + transPFC2 + rain+rain1	5.36	
transPFC1 + transPFC2+rain	1.55	null	5.39	
transPFC1 + rain	2.30	rain	5.47	
transPFC1 + rain1	3.14	transPFC1 + transPFC2+density+rain	5.49	
transPFC1 + density	3.14	transPFC1 + rain+rain1	5.79	
transPFC1 + transPFC2 + rain1	3.31	transPFC1 + density+ rain	5.79	

The mean PFC ± SD was 0.40 ± 0.24. For 3–11-year moving averages, the means were 0.40 with SEs decreasing monotonically from 0.063 to 0.016 with the window size of the moving average. Since the larger the window over which one computes a moving average the fewer moving averages result, we chose the smallest window that reflected the (partial) autocorrelations and the AR modelling results. Figure 3 together with the dominance of the first lag in the AR modelling suggested that a three-year moving average of PFC values would suffice for our data, employment of which did not alter the mean but reduced the SE to only 26% that of the raw data.

Discussion

For our study population, AR-modelling demonstrated negative correlations between PFC and its values in the previous two years, that with the previous year’s being the stronger, consistent with Fig. 3 and the mean length of inter-birthing intervals of 29 months. With a gestation of 15 months, a female that calved this year is extremely unlikely to have calved last year, and on average not the year before that either. Thus, the AR-modelling revealed the expected autocorrelation in the time series of PFC data. Total rainfall during the year in which PFC was measured was the most important of the three physical covariates but its relative importance was only 56% that of PFC lagged by two years and only 30% of that of PFC lagged by one year. Its regression coefficient is negative as is the correlation between the rain and PFC values (−0.186), counterintuitive to expectations of how rainfall might influence PFC. Density was the least important covariate. Since our detailed study of demography (Law, Fike & Lent, 2013; Law, Fike & Lent, 2014) detected no influence of relevant measures of rainfall on vital rates and birth sex and only density feedback on age at first reproduction, we conclude that for our study population the AR-modelling provided no false signal of density feedback but weak evidence of an influence of rainfall that is not supported by our more detailed prior studies (Law, Fike & Lent, 2013; Law, Fike & Lent, 2014). The success of AR-modelling in rejecting density as an influential covariate, no doubt reflects the fact that density increased monotonically during the study while no correlated trend in PFC values is apparent. As one would expect on statistical grounds, AR-modelling may, however, be misled to attribute variation in a time series of PFC values to a covariate (here rain) that exhibits some correlation with the PFC values even if it is not in fact driving that variation. Thus, our hypothesis is not unambiguously confirmed. Rather, when AR-modelling indicates the influence of a covariate on PFC values, independent evidence for an influence of that covariate on vital rates should be sought for confirmation. Such will also provide deeper insight into the dynamics of the population.

Discounting the physical covariates in the AR-modelling, the explained variation, manifested in the lagged versions of transPFC, can therefore be understood as due to the contingencies of breeding, including demographic-founding effects, in combination with inter-birthing intervals longer than a year, consistent with expectations. However, the global model left unexplained 54% of variation in transPFC. The unexplained variation left in age at first reproduction and inter-birthing intervals in Law, Fike & Lent (2013) and birth sex in Law, Fike & Lent (2014), was traced to the variation in birth numbers and sexes unexplained by the demographic models in Law, Fike & Lent (2015) and linked there to demographic stochasticity. We therefore propose that the variation in PFC unexplained by the AR modelling here also reflects demographic stochasticity.

PFC would appear not to be sensitive enough to detect the subtle signal of density feedback our individual-level demographic study detected; namely, an increase of age at first reproduction but no influence on inter-birthing intervals, birth sex, or population growth rate. This finding is similar to density feedback on conception rate in a population of white-tailed deer (Odocoileus virginianus borealis) despite none on population growth rate (McCullough, 1979:155) and consistent with expectations that density will impact juvenile/subadult fecundity (i.e., age at first reproduction) before impacting adult fecundity or survival in large ungulates, especially megaherbivores (Eberhardt, 2002). Detecting increasing age at first reproduction could warn of impending density feedback on population growth rate itself and therefore would be more useful than tracking PFC as a detector of density, even if more demanding of data collection. Indeed, if the per capita rate of population growth rate is ramp-like (i.e., exponential growth followed by a rapid decline to zero, (McCullough, 1992), (Owen-Smith, 2010) and Eberhardt’s schedule for the impact of density on vital rates is correct, one might expect PFC to be largely insensitive to density during much of the growth phase of a population of megaherbivores. But only further study of suitable populations can inform this issue. Indeed, the impact of density on megaherbivore population dynamics and vital rates remains an important gap in our knowledge, due to the rarity of long-term, detailed, studies of undisturbed growing populations.

Setting aside the question of employing PFC to study trends in population–level reproductive performance, even for a healthily expanding population such as our study population, PFC can vary considerably from year to year (Fig. 2), and therefore perhaps mislead as regards population reproductive performance if annual values are interpreted naively. While fluctuations in PFC will not be surprising when the population is small, even in 2007, ten years after introductions had ceased, PFC was only 0.26, very poor by du Toit’s benchmarks, yet 0.56 the year before and 0.45 the year after. Even more extreme, PFC in 2000 was 0.08, preceded by a value of 0.44 in 1999 and followed by 0.65 in 2001. Very high and very low PFC values are to be expected with even moderately sized populations of 50–100, especially when demographic-founding effects persist. The introduced females in our study population consisted of 71% subadults, of which 70% were introduced in 1997. Coincident first reproductions of introduced females were a component of demographic-founding effects in our study population that contributed to high PFC values in some years and low values in the succeeding year. But variation in both inter-birthing intervals and birth sex continued in our study population through 2008 even as the population stage structure closely approached a stable distribution. This variation was attributed to demographic stochasticity in Law, Fike & Lent (2015) and must be a component of the fluctuations in PFC for our study population. Consistent with our analyses and expectations of no environmental influence and only a subtle density feedback, there is no apparent trend in PFC values or a sustained reduction to lower values at any point. There was also considerable variation about a trend in PFC values detected by Hrabar and du Toit (2005; Fig. 4) and in Okita-Ouma’s (2014; Fig. 3.4) data for seven populations, with examples of no trend, increasing trend, and decreasing trend against (delayed) density (see also Okita-Ouma Fig. 5.5 for plots of three-year moving averages).

Moving averages can expose longer term trends by suppressing shorter term fluctuations. PFC values for our study population were never consistently at least 0.4, the threshold for du Toit’s benchmark for good to excellent rating. Moving averages converged on the mean PFC value with increasing window size (consistent with an absence of any linear trend in the data over time). For a fixed window, the overall mean was not altered but the SD decreased, by 74% even for a three-year moving average. Moving averages could be helpful in evaluating the significance of individual PFC values, e.g., when applying a rule of thumb like du Toit’s to monitor introduced rhinoceros populations. Partial autocorrelations in time series of PFC values might also be driven by environmental stochasticity, however, and should not be simply averaged away without discretion.

Conclusion

We have presented a study, with high quality data, of accurate PFC values for a population of black rhinoceros. Our aim was not to advocate for or against PFC as a useful surrogate in the absence of individual-level demography but to test its performance on our study population. A time series of PFC values for a population of megaherbivores can be expected to exhibit autocorrelations reflecting inter-birthing intervals longer than a year thus causing fluctuations and to also manifest variation arising from founding effects and demographic stochasticity at population sizes not atypical for megaherbivores. AR modelling may misinterpret covariates as drivers of such variation in PFC when merely correlated, though the evidence provided for such a false signal by the model may be only weak. Evidence from analysis of PFC values for drivers of population-level reproductive performance should therefore be supplemented by more detailed study of the potential influence of a covariate on actual vital rates. PFC also appears less sensitive than age at first reproduction at detecting incipient density feedback. In the absence of driven fluctuations, PFC can still show considerable variation, especially for introduced populations, for at least a couple of decades. During this period, moving averages of PFC may reflect population-level reproductive performance better than PFC itself. Further testing of PFC in other populations would be valuable, especially those exhibiting density feedback on vital rates. We conclude that while PFC may be a convenient tool to attempt to measure population-level reproductive performance, one must be careful not to over interpret its variation. For megaherbivores at least, it should perhaps be regarded as a preliminary measure that can indicate the direction of more intensive demographic study.

Supplemental Information

Data S1 Raw data

Click here for additional data file.

We thank the referees for useful suggestions that improved the manuscript and Dean Peinke at the Eastern Capes Park and Tourism Agency.

Additional Information and Declarations

Competing Interests

Author Contributions

Data Availability

The authors declare there are no competing interests.

Peter R. Law conceived and designed the experiments, performed the experiments, analyzed the data, prepared figures and/or tables, authored or reviewed drafts of the paper, approved the final draft.

Brad Fike conceived and designed the experiments, performed the experiments, contributed reagents/materials/analysis tools, authored or reviewed drafts of the paper, approved the final draft.

The following information was supplied regarding data availability:

The raw data are provided in Data S1.

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
