# Peer review of "Testing ‘Proportion of Females Calving’ as an indicator for population-level reproductive performance for black rhinoceros (Diceros bicornis)"

_PeerJ, doi:10.7717/peerj.5430_

## Round 0.1 · original submission · Major Revisions

The reviewers found your manuscript valuable, but indicated a number of concerns. Please revise it, paying close attention to all of the comments of the reviewers.

# Reviewer 1 ·

Basic reporting

1. The manuscript was generally well-written using professional English. I believe, however, there was an excessive use of non-standard acronyms (e.g., PFC, IBI, AFR, ACF) that made for difficult reading, and required flipping back and forth to find the meaning of particular acronyms.
2. The manuscript provided a reasonable background with appropriate literature citations. I was surprised that the authors didn’t use the IUCN status of black rhinoceros (Critically Endangered, but with increasing populations) as a justification for their research.
3. The manuscript conformed to PeerJ standards.
4. The figures were clear, but I didn’t see the need to provide the transformations along with the original data in Fig. 1, which cluttered the figure. Also the Y-axis on Fig. 1 should be labeled. I believe it would be helpful to present the Percent of Females Calving in conjunction with yearly data on population size in the same figure (data on population size are only available in the supplemental data). The time-series analysis in Fig. 2 was inventive, but the levels of significance represented by the horizontal dashed lines needs to be identified.
5. Raw data were provided in a table.

Experimental design

1. This original research falls within the scope of the Journal. The authors are to be commended for their long data run, which is unusual for large mammals.
2. The research question was relevant; there are few data on asynchronous non-seasonable breeders with an inter-birth interval of more than 1 year. A major shortcoming, however, was the failure to set forth a set of hypotheses at the end of the Introduction and the criteria necessary to test them. Framing the Discussion around those hypotheses also would improve the organization of that section, and more clearly indicate how the knowledge gap would be filled by this study.
3. The research was rigorous and likely performed to high ethical standards, although I would like to have seen an animal care and use approval (or necessary permits) for the original research, even though the authors were just analyzing data in this paper. The Methods were clear, but the authors relied heavily on previous publications, and providing a bit more information would have been useful.

Validity of the findings

1. There are several ways to use AIC modeling. I wonder what the analysis would have looked like if only the top 3 models (those with changes in AIC of less than 2 from the top model) were averaged. The influence of precipitation is discounted, but rain occurs in one of those top models.
2. Some evidence of density dependence was reported in their previous publications—a relationship occurred between age at first reproduction and population density. The critical point is not density per se, but density in relation to ecological carrying capacity (K). Parametrizing where the population is with respect to K is necessary to see how percent of females calving might change as the population approaches K. I doubt there are data in this manuscript to allow that assessment, but the Discussion would be improved by a more complete discussion of ungulate population dynamics and some predictions about how things might change in the future as density continues to increase.

Additional comments

1. Long data runs of this type are unique and critical to our understanding of population dynamics and metrics that reflect the status of populations.
2. A clear-cut approach to stating and testing hypotheses would provide a more coherent paper with less ambiguous conclusions.
3. Placing this research in a firmer context of ungulate population dynamics would improve the scope of the paper and help with identifying gaps in our knowledge.

Reviewer 2 ·

Basic reporting

This manuscript addresses a specific aspect of population monitoring using a detailed and impressive data set from a reintroduced population of black rhinos. As such, the work is likely useful for managers of populations in rhinos.

The language is generally clear and concise, however, there are several aspects of the organization and writing that would benefit from revision.
1. The title does not provide a reader with a clear understanding of the content of the paper without reading at least the Abstract because PFC is not a common abbreviation.
2. Similarly, although the population name (SKKR) is “not an acronym of an official name”, it is distracting, leaving the reader to wonder what it does stand for.
3. Figure and Table legends and captions are too vague/brief to allow readers to understand the data presented and its significance. More context would help with interpretation of the information presented. For example, in Table 1 the abbreviations should be identified either in the caption or in a footnote.
4. The Introduction could do a better job of setting up the problem and providing context for why this topic should be of interest to a broad readership. The topic is so narrow that it comes across almost as a rebuttal than a presentation of research results in a stand along article. This is apparent when discussing benchmarks for populations (Lines 50-54 and again in the Discussion in the paragraph beginning with line 176).
5. A large number of statements/conclusions are presented in the Introduction without data or citations for where the data could be found (e.g., statements in lines 63-71 and 81-83). This also contributes to the feeling that this short paper is an addendum rather than a complete study or analysis.

A few specific comments/suggestions for improvement:
Line 128: This paragraph could use some context and a strong topic sentence.
Line 134: This statement and citation are more appropriate for Discussion than Results.
Line 137-138: This information and citation belong in the Methods. The AIC weights would be better presented in table.
Line 144-146: What criteria was used to make this decision?

Experimental design

Data on population size, sex ratios, reintroduction history, and stage structure over the study period would help with interpretation of the analyses. The reliance on previously published papers for this information requires that readers trust the statements without an opportunity to evaluate them.

Validity of the findings

The data and analyses are relatively simple, however, there are a couple of points regarding interpretation that could use additional thought.

1. The sample size of adult females is quite small during the study period (<8 individuals for the first half). The high stochastic variation in reproduction during the early time period should not be surprising given the small number of individuals, and the reduction in variation overtime likely reflects, in part, an increase in sample size. This temporal trend should be discussed/acknowledged in the Discussion, and it might be more transparent to include the numbers of adult females in the text or perhaps in the figure legend (that information is included in the supplementary table).
2. A related comment regarding interpretation is that the long intervals between births (~2.5 yr) should result in the lag effects in reproduction given even a modest amount of synchrony among females. It almost would be surprising if this was not the case.
3. Density is not important in this population yet because it is not close to carrying capacity. How does this influence interpretation of your results?

---

## Round 0.2 · Minor Revisions

Please make the minor revisions requested by Reviewer 1. Note that data should be plural throughout. I advise you to use the reviewer's suggestion on ethical guidelines for research on wild mammals.

Reviewer 1 ·

Basic reporting

The hypothesis testing was improved.

Experimental design

The authors satisfied my concerns with their analysis.

Validity of the findings

I thought the authors did a better job of casting findings from their research in a broader framework that acknowledged the potential value of their findings.

Additional comments

The authors have done a reasonable job of revising their manuscript, and satisfied most of my concerns.

There remain a few small corrections:
Lines 100-105—break this long sentence into several clear sentences.
Lines 110, 139—data are plural and these should read “data are” not data is.

There is one point, however, that I would like to see resolved. The authors point out that data used in their manuscript were collected as a result of management objectives rather than a research project. I do not believe that his absolves them, however, from indicating that data were collected in a responsible manner. The manuscript is research and it is their responsibility to indicate whether data they used were collected in a conscientious fashion reflecting the ethical treatment of animals. Perhaps a way to accomplish this would be to indicate that data were collected in keeping with guidelines for research on wild mammals (Sikes et al. 2011, Journal of Mammalogy 92:235-253).

Reviewer 2 ·

Basic reporting

Revision addresses comments.

Experimental design

Revision addresses comments.

Validity of the findings

Revision addresses comments.

Additional comments

Although the rebuttal has a somewhat argumentative approach, the authors have addressed my concerns and comments sufficiently.

---

## Round 0.3 · accepted · Accept

Thank you for making those changes, and congratulations on a nice paper!

#